# The Effects of Breed, Lactation Number, and Lameness on the Behavior, Production, and Reproduction of Lactating Dairy Cows in Central Texas

Lily A. Martin [1], Edward C. Webb [1,*], Cheyenne L. Runyan [1], Jennifer A. Spencer [2], Barbara W. Jones [3] and Kimberly B. Wellmann [1]

1　Department of Animal Sciences, Tarleton State University, Stephenville, TX 76402, USA; lily.martin@go.tarleton.edu (L.A.M.); runyan@tarleton.edu (C.L.R.); kwellmann@tarleton.edu (K.B.W.)
2　Texas A&M AgriLife Extension, Stephenville, TX 76402, USA; jennifer.spencer@ag.tamu.edu
3　Department of Agriculture, Eastern Kentucky University, Richmond, KY 40475, USA; barbara.jones@eku.edu
*　Correspondence: ewebb@tarleton.edu

**Simple Summary:** Producers maintain cows in their herd based on breed type, lactation number, and individual productivity characteristics. However, there are questions about the effect of management decisions regarding cow behavior and productivity on cow welfare and the economic viability of dairy farms in Central Texas, which is not an ideal environment for dairy production. Data from this study confirm that breed and lactation numbers influence milk yield, milk urea nitrogen concentrations, lying time, and the incidence of lameness. Also, a higher lameness incidence was found in high-producing cows, with lower milk urea nitrogen concentrations. It is suggested that management decisions regarding the retention of cows, breed types, and lactation number influence the incidence of lameness, with subsequent effects on cow behavior and productivity.

**Abstract:** The objective of this study was to evaluate the effects of breed, lactation number, and lameness on lying time, milk yield, milk urea nitrogen concentration (MUN), progesterone concentration ($P_4$), and the calving-to-conception interval (CCI) of lactating dairy cows in Central Texas. A total of 84 lactating dairy cows (Holsteins, Jerseys, and crossbreeds) from a commercial dairy farm in Central Texas were randomly selected and enrolled in this study from October 2023 to February 2024. Cows ($60 \pm 7$ DIM) were enrolled in cohorts weekly for five weeks and were randomly fitted with an IceQube pedometer (IceRobotics, Edinburgh, UK) to track lying time. Lameness and body condition scores (BCS) were recorded, and blood samples were collected once a week. Parameters of reproductive performance included insemination rate, conception rate, pregnancy rate, and the CCI. Monthly dairy herd improvement association (DHIA) testing included milk yield and MUN concentrations. Breed and lactation number had a significant effect on milk yield, MUN concentration, lying time, BCS, and lameness ($p < 0.001$). Lactation number had a significant effect on $P_4$ concentrations ($p < 0.001$). There was a positive correlation between lameness and milk yield ($p = 0.014$) and a negative correlation between lameness and MUN concentrations ($p = 0.038$).

**Keywords:** behavior; breed; lactation number; lameness; production; reproduction

## 1. Introduction

There are various factors to consider when evaluating production performance in dairy cows. This includes the possible relationships between lameness and lying time and their effect on production, such as milk yield and reproductive performance. Lameness, which is defined as a change in gait due to the cow alleviating pain in the limbs [1], negatively affects behavior by altering lying time, social interactions, estrus intensity, feeding, and rumination behaviors, which makes it a major welfare concern [2]. Lameness causes an increase in lying time due to the assumed pain in switching positions [3–6] and negatively impacts

milk yield [3,7,8] and reproduction [9–11]. Thus, lameness is one of the top three health problems causing premature culling [12] and is the third-most important health-related cost behind fertility and mastitis [3].

The average lying time among dairy cows is 11 h/d, but varies among cows from less than 6 h/d to more than 16 h/d [5,13,14]. Factors associated with lying time include parity, DIM, body condition score (BCS), lameness, and milk yield [5,15]. Management decisions also impact lying time, which can include stall design, bedding material, stocking density, environmental conditions, and milking and nutrition management [16–18]. The use of precision dairy farming technologies has become popular among dairy producers due to the ability of automated measurements. These technologies provide information about behaviors such as lying time, potential diseases and lameness, and rumination time [19–21]. The use of activity measurements (i.e., lying behaviors, time spent standing, number of steps) can aid in detection of estrus [13,22]. However, limited research suggests that an increase in lying time is associated with reduced reproductive performance [23].

Assessing reproductive performance includes measurements such as insemination rate, conception rate, pregnancy rate, and the calving-to-conception interval [10]. Insemination rate is defined as the number of animals inseminated divided by the number of animals eligible to be inseminated in a given time frame multiplied by 100 [10]. Conception rate is defined as the number of known pregnant animals divided by the number of inseminated animals in a given time frame multiplied by 100 [10]. Pregnancy rate is defined as the number of known pregnant animals divided by number of cows eligible to become pregnant in a given time frame multiplied by 100 [10]. If a cow does not become pregnant, progesterone concentrations will begin to decline [24], thus allowing the use of progesterone concentrations to determine ovarian cyclicity [25].

Milk yield has been associated with reproductive performance, where a longer calving-to-conception interval causes an increase in days until the cow can be milked [26]. However, there are other factors to consider when evaluating milk yield, including variations between breeds, parity, days in milk (DIM), postpartum metabolic disorders, milking management, nutrition, and milk components. Research has shown a negative correlation between lying time and milk yield [23,27], suggesting that a decrease in lying time is due to a longer time eating to maintain high milk production. However, lameness causes a decrease in milk yield due to cows lying down longer [3,7,8] and thus eating less. Additionally, nutrition, through effects on milk urea nitrogen (MUN) concentrations, influences the intercalving period and reproductive efficiency of dairy cows [28].

There is a lack of research on the relationship and effects of breed, lactation number, and lameness on lying time, milk yield and composition, and reproductive performance. Therefore, assessing these behaviors with milk production and reconception rates may provide further insight into improving farm management. The researchers hypothesized that breed, lactation number, and lameness will affect behavior, milk production, and reproductive performance. The objective of this study was to evaluate the effect of breeds, lactation numbers, and incidence of lameness on lying time, milk yield, and reproductive performance in lactating dairy cows starting at the end of the voluntary waiting period to allow for complete postpartum involution of the reproductive system.

## 2. Materials and Methods

### 2.1. Study Design

Lactating dairy cows (N = 84) located on a commercial dairy farm in Central Texas were recruited for this study from October 2023 to February 2024. Cohorts of cows (60 ± 7 DIM) were entered in the study weekly for five consecutive weeks. Cows differed in breed (Holstein = 22, Jersey = 37, and crossbreed = 25) and lactation number (1st = 6, 2nd = 38, 3rd = 14, 4th = 14, and 5th = 12).

*2.2. Facilities and Nutrition Management*

Cows were housed in naturally ventilated free-stall barns with sand bedding, grooved alleyways, and fans placed above pens. Bedding material was raked before each milking and replenished with sand once a week. Holsteins and Jerseys were housed in separate pens and crossbreeds were housed throughout the pens based on size. Stall dimensions for Jersey pens included the width of individual stall (1.2 m), length of stall to the curb (2.5 m), length of stall from the brisket board to the curb (1.8 m), and height from the neck rail to the bed (1.2 m). The same measurements were taken for the Holstein pens and measured 2 m larger than the Jersey pens, because of the larger size of Holsteins. Manure management consisted of a manure vacuum and scraper that went through the barns every 2 h. The milking parlor consisted of a 30-stall herringbone parlor and cows were milked 3 times per day. Monthly on-farm milk testing was performed through the Dairy Herd Improvement Association (DHIA). A foot bath containing a formaldehyde and copper sulfate solution was placed outside of the milking parlors before cows returned to their pens. Cows were evaluated once a week for hoof trimming procedures. Cows were fed every 2 h from 0900 h to 1700 h. Cows had ad libitum access to water.

*2.3. Reproductive Management*

The farm utilized natural heat detection and CowManager SensOor ear tags (Agis Automatisering BV, Harmelen, The Netherlands) to identify cows in heat. On-farm reproductive management consisted of three artificial insemination (AI) technicians and two veterinarians. Veterinarians conducted transrectal ultrasonography every two weeks to diagnose conception rates in cows (35 to 45 d post-AI). Cows were either classified as pregnant, open, or having no significant structure (corpus luteum (CL) or ovarian follicles). If cows were open with a CL circa 15 mm, then an Estrumate injection (2 cc; Merck Animal Health, Merck & Co., Inc., Rathway, NJ, USA) was administered intramuscularly to cause regression of the CL. If cows were diagnosed pregnant, they were reevaluated at 60 d post AI to assess pregnancy rates.

*2.4. Data Collection*

Individual cow data including DIM, breed, lactation number, previous calving date, breeding date(s), number of services, veterinary reproduction checks, average milk yield, milk urea nitrogen (MUN) concentration, and health events (sick, lame, mastitis, hoof health) were obtained from on-farm records using PCDart (Dairy Records Management Systems; DHI Cooperative Inc., Columbus, OH, USA). There were no exclusions based on the number of services or health events. Average milk yield and MUN concentration were provided through monthly DHIA tests. Temperature (F°) and relative humidity (RH, %) were recorded each day with a HOBO data logger (Onset, Bourne, MA, USA). The temperature humidity index (THI) was calculated as follows:

$$THI = 0.8 \times T + RH/100 \times (T - 14.4) + 46.4$$

where T = ambient daily temperature in °C and RH = ambient daily relative humidity [29].

Cows were randomly assigned to an IceQube pedometer (Peacock Technology Ltd., Scotland, UK) that was fit to their left rear leg during enrollment and remained on the leg throughout the duration of the study. Data from the IceQubes consisted of lying time, lying bouts, time spent standing, and steps. Pedometers were scanned with electronic readers once a week to collect data from individual cows. The data were exported from the IceManager software Version 6.1 into an Excel file. Lying time data were summarized and reported daily. Body condition scoring (1–5-point scale with 0.25 increments; Elanco Animal Health) and lameness scoring (1–5-point scale; a score of 1 being non-lame and a score of 5 being severely lame) [12,30] were conducted once a week by one trained individual throughout the study. Lameness scoring began one week after fitting cows with IceQubes due to acclimation.

Blood samples were collected via coccygeal venipuncture from the tail head into a 10 mL BD vacutainer. Blood samples were centrifuged and serum was extracted into 1.5 mL microcentrifuge tubes and stored at $-80\,^{\circ}$C until transportation to an off-site lab for progesterone ($P_4$) analysis. To evaluate the reproductive performance within the study population, various parameters were used, including insemination rate, conception rate, calving-to-conception interval (CCI), and pregnancy rate.

### 2.5. Statistical Analysis

All statistical analyses were performed using IBM$^{®}$ SPSS$^{®}$ version 29.0. Data of individual cows, including breed, lactation number, health events, previous calving, and number of inseminations, were exported from the producer's computer system, PCDart, into Excel. A general linear model appropriate for repeated measures was used to evaluate lying bouts (frequency), lying time (duration), $P_4$, CCI, average MUN concentration, average milk yield, DIM, BCS, lameness, and THI by the main effect of breed and lactation number. The cow served as the experimental unit. Post hoc Bonferroni adjustment was used to compensate for an unbalanced design in terms of the number of cows for different breed types. Post hoc tests were not performed on lactation number, since there were fewer than two cases in the first lactation among cows that conceived. Pearson's chi-squared analysis was used for the analyses of categorical data, namely, BCS and lameness scores. The interaction effects between breed and lactation number were also assessed. Finally, a bootstrap analysis was conducted to obtain correlations with lactation numbers. Alpha was set at 0.05; therefore, $p \leq 0.05$ was considered significant, and tendencies were discussed at $0.05 < p \leq 0.10$.

## 3. Results

### 3.1. Milk Yield and MUN Concentrations

Total mean $\pm$ SEM for DIM, milk yield, and MUN concentrations varied by breed and lactation number (Table 1). There were no differences in DIM among breeds (Holsteins = $99 \pm 25.9$, Jersey = $97 \pm 24.3$, and crossbreeds = $99 \pm 24.4$ DIM); however, there was a difference in DIM by lactation number ($p = 0.003$). Holsteins yielded $11.2 \pm 0.76$ and $7.5 \pm 0.83$ kg more milk compared to Jerseys and crossbreeds, respectively ($p < 0.001$). Jerseys had the highest MUN concentrations compared to Holsteins and crossbreeds ($p < 0.001$ and $p = 0.006$, respectively), but no significant differences were found between Holsteins and crossbreeds.

**Table 1.** The effects of breed and lactation number on mean number of days in milk, milk yield, and milk urea concentration of dairy cows included in the study.

| Breed | Days in Milk [1] | Milk Yield [1], kg | Milk Urea N [1], mg/dL |
|---|---|---|---|
| Holstein | $99 \pm 25.9$ | $40.10 \pm 10.056$ [a] | $14.53 \pm 4.008$ [a] |
| Jersey | $97 \pm 24.3$ | $28.90 \pm 7.082$ [b] | $16.54 \pm 4.139$ [b] |
| Crossbreed | $99 \pm 24.4$ | $32.58 \pm 8.590$ [c] | $15.31 \pm 4.143$ [a] |
| **Lactation number** | | | |
| 1 | $98 \pm 22.2$ | $27.44 \pm 6.005$ [a] | $16.85 \pm 4.725$ [a, c] |
| 2 | $96 \pm 22.8$ | $33.38 \pm 8.323$ [b] | $16.44 \pm 4.09$ [a] |
| 3 | $103 \pm 27.5$ | $33.04 \pm 10.574$ [b] | $14.39 \pm 4.322$ [b, c] |
| 4 | $103 \pm 27.5$ | $37.93 \pm 9.776$ [c] | $15.33 \pm 3.769$ [a–c] |
| 5 | $92 \pm 21.1$ | $28.76 \pm 9.518$ [a] | $14.84 \pm 3.916$ [a–c] |

[1] Least square means $\pm$ SEM; [a, b, c] means with different superscript letters in the same column and within a factor (breed and lactation number) differed ($p < 0.001$).

Lactation number had a significant effect on milk yield ($p < 0.001$). As expected, cows in their first lactation had lower milk yield compared to cows in their second, third, and fourth lactations ($-5.9 \pm 1.26$ kg, $-5.6 \pm 1.34$ kg, and $-10.5 \pm 1.39$, respectively). Cows in their fourth lactation had greater milk yield compared to cows in their second and third lactation ($4.6 \pm 0.91$ kg and $4.9 \pm 1.02$ kg, respectively). Cows in their fifth

lactation had lower milk yield compared to cows in their second, third, and fourth lactation ($-4.6 \pm 0.97$ kg, $-4.3 \pm 1.07$, and $-9.2 \pm 1.12$ kg, respectively). Lactation number had a significant effect on MUN concentrations ($p < 0.001$). Cows in their first lactation had higher MUN concentrations compared to cows in their third lactation ($2.46 \pm 0.713$, $p = 0.009$). Cows in their second lactation had higher MUN concentrations compared to cows in their third and fifth lactation ($2.04 \pm 0.448$; $p < 0.001$ and $1.6 \pm 0.511$; $p = 0.028$, respectively).

Cows That Conceived

In total, 56 cows conceived and were subsequently analyzed for DIM, milk yield, and MUN concentrations (Table 2). Breed had a significant effect on milk yield, where crossbreeds yielded more milk than Jerseys ($p = 0.03$). No differences in milk yield were found between Holsteins and the other breeds. Lactation number tended to affect milk yield ($p = 0.089$), where third-lactation cows tended to produce more milk than cows in their second, fourth, and fifth lactation. There was a significant negative correlation between milk yield and MUN concentrations ($r = -0.481$; $p = 0.027$). Thus, as milk yield increased, MUN concentrations decreased.

**Table 2.** The effect of breed and lactation number on mean days in milk, milk yield, and milk urea nitrogen for cows that conceived.

| Breed | Days in Milk [1] | Milk Yield [1], kg | Milk Urea N [1], mg/dL |
|---|---|---|---|
| Holstein | $112 \pm 14$ | $36.83 \pm 8.449$ [a, b] | $17.75 \pm 2.5$ |
| Jersey | $121 \pm 17$ | $29.23 \pm 7.316$ [b] | $19.31 \pm 3.521$ |
| Crossbreed | $127 \pm 16$ | $33.24 \pm 9.404$ [a] | $17 \pm 2.345$ |
| Lactation number | | | |
| 2 | $125 \pm 18.3$ | $32.76 \pm 7.689$ [c] | $18.85 \pm 3.693$ |
| 3 | $109 \pm 3.8$ | $43.08 \pm 13.411$ [d] | $15.67 \pm 1.528$ |
| 4 | $119 \pm 17.4$ | $32.73 \pm 8.822$ [c] | $18.33 \pm 1.528$ |
| 5 | $112 \pm 11.3$ | $23.93 \pm 12.477$ [c] | $19.50 \pm 2.121$ |

[1] Least square means $\pm$ SEM. [a, b] means with different superscript letters in the same column and within a factor (breed and lactation number) differed ($p = 0.03$); [c, d] means with different superscript letters in the same column and within a factor (breed and lactation number) tended to differ ($p = 0.089$).

### 3.2. Body Condition, Lameness, and Lying Time

Body condition score (BCS), lying time, and lameness differed between breeds and lactation numbers (Table 3). Jerseys had a higher BCS ($p < 0.01$) compared to Holsteins, but no differences were found from crossbreeds. Lactation number had a significant effect on BCS ($p < 0.01$). Results suggest that as lactation number increases, BCS decreases.

**Table 3.** The effect of breed and lactation number on mean body condition score, lameness, and lying time throughout the study.

| Breed | Body Condition Score [1] | Lameness [1] | Lying Time [1] |
|---|---|---|---|
| Holstein | $3.0 \pm 0.24$ [a] | $1.3 \pm 0.51$ [a] | 8:40 $\pm$ 2:39:11 [a] |
| Jersey | $3.1 \pm 0.25$ [b] | $1.1 \pm 0.42$ [b] | 9:35 $\pm$ 2:37:34 [b] |
| Crossbreed | $3.1 \pm 0.24$ [a, b] | $1.3 \pm 0.67$ [a] | 8:43 $\pm$ 2:21:59 [a] |
| Lactation number | | | |
| 1 | $3.2 \pm 0.26$ [a] | $1.1 \pm 0.22$ [a] | 8:05 $\pm$ 2:41:16 [a] |
| 2 | $3.1 \pm 0.22$ [a, b] | $1.1 \pm 0.41$ [a] | 8:58 $\pm$ 2:32:35 [a] |
| 3 | $3.1 \pm 0.23$ [b] | $1.1 \pm 0.25$ [a] | 8:59 $\pm$ 2:05:29 [a] |
| 4 | $2.9 \pm 0.25$ [c] | $1.7 \pm 0.74$ [b] | 8:57 $\pm$ 2:25:44 [a] |
| 5 | $3.0 \pm 0.27$ [d] | $1.2 \pm 0.59$ [a] | 10:13 $\pm$ 3:11:18 [b] |

[1] Least square means $\pm$ SEM. [a–d] means with different superscript letters in the same column and within a factor (breed and lactation number) differed ($p < 0.01$).

The effect of breed on lameness was significant ($p < 0.01$). Jerseys had lower lameness scores compared to Holsteins and crossbreeds. Lactation number also influenced lameness significantly ($p < 0.001$). Results suggest that cows in their fourth lactation had higher lameness scores than lactations one through three and lactation five.

Jerseys had an average lying time of $9.5 \pm 2.75$ h/d, Holsteins had an average lying time of $8.5 \pm 2.75$ h/d, and crossbreeds had an average lying time of $8.75 \pm 2.25$ h/d. There were no differences in lying time between Holsteins and crossbreeds. Lactation number had a significant effect on lying time, where cows in their fifth lactation laid down longer than cows in other lactations ($p < 0.01$).

Cows That Conceived

Cows that conceived during the trial were analyzed for BCS, lameness scores, lying duration, and lying bouts (Table 4). Among lactations, there was a significant positive correlation between lameness and milk yield ($r = 0.525$, $p = 0.014$) and a significant negative correlation between lameness and MUN concentrations ($r = -0.448$, $p = 0.042$). Breed had a significant effect on lying bouts ($p = 0.038$). Jerseys had more lying bouts compared to Holsteins and crossbreeds, with Holsteins having the least fewest lying bouts. The mean lying bout frequency for cows that conceived was $7 \pm 2.5$ lying bouts/d. Lying time and lying bouts had a significantly positive correlation ($r = 0.461$; $p = 0.035$).

**Table 4.** The effect of breed and parity on mean body condition score, lameness score, lying time, and lying bouts for cows that conceived.

| Breed | Body Condition Score [1] | Lameness [1] | Lying Time [1] hh:mm:ss | Lying Bouts [1] |
|---|---|---|---|---|
| Holstein | $2.9 \pm 0.31$ | $1.3 \pm 0.50$ | 9:35 ± 3:32:00 | $6.0 \pm 1.41$ [a] |
| Jersey | $3.0 \pm 0.30$ | $1.0 \pm 0.00$ | 8:39 ± 2:10:52 | $7.5 \pm 2.63$ [b] |
| Crossbreed | $3.1 \pm 0.14$ | $1.6 \pm 0.55$ | 7:50 ± 2:44:25 | $6.4 \pm 2.79$ [a] |
| Lactation number | | | | |
| 2 | $3.1 \pm 0.22$ | $1.1 \pm 0.28$ | 8:15 ± 2:08:54 | $6.9 \pm 2.28$ |
| 3 | $3.0 \pm 0.25$ | $1.7 \pm 0.58$ | 9:32 ± 0:57:06 | $7.3 \pm 1.53$ |
| 4 | $2.9 \pm 0.38$ | $1.7 \pm 0.58$ | 8:54 ± 4:28:37 | $6.0 \pm 0.00$ |
| 5 | $2.6 \pm 0.18$ | $1.0 \pm 0.00$ | 8:27 ± 3:33:02 | $7.0 \pm 2.49$ |

[1] Least square means ± SEM. [a,b] means with different superscript letters in the same column and within a factor (breed and lactation number) differed ($p < 0.05$).

### 3.3. Reproduction

Progesterone concentration, the number of times a cow was serviced, assuming insemination, and calving-to-conception interval varied (Table 5). The average CCI was 122 d for all breeds. Reproductive measurements for insemination rate, conception rate, and pregnancy rate were 95%, 76%, and 72%, respectively. The relationship between $P_4$ concentrations and CCI tended to have a positive correlation ($r = 0.418$; $p = 0.059$). The relationship between $P_4$ concentrations and THI had a significant negative correlation ($r = -0.432$; $p = 0.05$). The relationship between CCI and THI tended to have a negative correlation ($r = -0.42$; $p = 0.058$).

**Table 5.** Mean circulating progesterone, total number of inseminations, and calving-to-conception interval (CCI) for cows that conceived during the trial.

| Breed | Cows Conceived | Insemination [1] | Progesterone [1] ng/mL | CCI [1] |
|---|---|---|---|---|
| Holstein | 4 | $1.25 \pm 0.50$ | $8.66 \pm 1.859$ | $110 \pm 14.1$ |
| Jersey | 13 | $1.38 \pm 0.65$ | $11.44 \pm 4.759$ | $121 \pm 17.6$ |
| Crossbreed | 5 | $1.40 \pm 0.55$ | $24.94 \pm 16.106$ | $126 \pm 15.8$ |

**Table 5.** *Cont.*

| Breed | Cows Conceived | Insemination [1] | Progesterone [1] ng/mL | CCI [1] |
|---|---|---|---|---|
| Lactation number | | | | |
| 2 | 13 | 1.54 ± 0.66 | 28.24 ± 41.59 | 125 ± 18.8 |
| 3 | 3 | 1.00 ± 0.00 | 5.26 ± 1.23 | 108 ± 5.5 |
| 4 | 3 | 1.00 ± 0.00 | 10.82 ± 3.45 | 119 ± 15.9 |
| 5 | 2 | 1.00 ± 0.00 | 11.44 ± 4.48 | 110 ± 12.7 |

[1] Least square means ± SEM.

## 4. Discussion

This study evaluated the effects of breed, lactation number, and lameness on the behavior, production, and reproduction of lactating dairy cows. No interactions were present in the results. Results from this study show that breed and lactation number affect milk yield, MUN concentrations, BCS, lameness, and lying time. Results from the current study also show lameness to affect milk yield and MUN concentrations.

### 4.1. Milk Yield and MUN Concentration

Similarly to the current study, other studies have also found that Holsteins have a greater milk yield compared to crossbreeds and Jerseys [31–33]. This is due to the higher genetic merit for milk yield in Holsteins compared to Jerseys and crossbreeds [31]. Throughout the study, first-lactation cows produced the least milk, then demonstrated an increase through the fifth lactation. The findings that lactation number had a significant effect on milk yield agrees with other studies: multiparous cows produce more milk than primiparous cows [34,35]. For example, one study found that cows in parity two and three produced 16.9% and 26.6% higher milk yield than parity-one cows, respectively [31]. This could be because primiparous cows are usually bred at about 24 months old and are still growing at this stage [34]. The incompletely developed mammary glands, combined with the demand for nutrients to complete the primiparous body's development, are the main factors responsible for the lower production in the first lactation [36].

According to our data for cows that conceived, cows in the third lactation yielded 43.08 ± 13.411 kg of milk, but the mean milk yield decreased to 32.71 ± 8.822 and 23.93 ± 12.477 for lactation number four and five, respectively. The difference in our findings compared to other research could be due to individual cow characteristics and management practices at the farm. Although no interaction effects were significant between breed and lactation number, because most Jerseys conceived, but had an overall lower milk yield than Holsteins and crossbreeds, this may explain the decrease in milk yield. A decrease in milk yield for later-lactation cows could also be due to cows having an increased chance of lameness, which was found in our study and agrees with other studies [31,37,38]. Additionally, the majority of the later-lactation cows had a BCS of 3, while cows in their fourth lactation had a lower BCS, which may explain the low milk yield for that group.

A negative correlation was observed between milk yield and MUN concentrations for cows that conceived during the experiment. This finding suggests that as the milk yield increased, MUN concentrations decreased. However, our study disagrees with recent research that suggests milk yield and MUN concentrations have a positive correlation [28]. Milk urea nitrogen increases for several months after parturition, then decreases and levels off during mid-lactation [39]. In later lactation, as milk production declines, the protein requirements decrease, suggesting a decline in MUN concentrations [40]. However, overfeeding protein during late lactation can cause MUN concentrations to be maintained during a decrease in milk yield [40], which may explain some of the variation observed in the present study. Therefore, proper nutrition is important in maintaining an appropriate MUN level throughout the dairy herd.

The correlation between milk yield and MUN concentrations could also be due to the Jerseys in our study having higher MUN concentrations and a low milk yield. However,

other studies have found Jerseys to have a low MUN concentration compared to Holsteins [28,41,42]. Research has found Holsteins average 15.2 mg/dL and Jerseys average 13.7 mg/dl, suggesting that MUN concentration is a function of body weight, with larger breeds having higher MUN concentrations [42]. Thus, larger breeds need to consume more feed to maintain their body weight compared to smaller breeds. However, another study found no differences in MUN concentrations between Jerseys and Holsteins [43], which agrees with cows able to conceive not having a difference in MUN concentrations between breeds. Researchers found that the crude protein concentration directly affects MUN concentrations [43]. Holsteins fed a diet with higher rumen degradable protein compared to Jerseys had higher MUN concentrations [28]. Thus, the differences among findings could be due to all breeds being fed the same TMR, thus leading to overfeeding Jerseys by the protein content.

In our study, we found MUN concentrations to be highest in first and second lactations versus cows in their third and fifth lactations. Thus, as parity increases, we can assume a decrease in MUN concentration within this herd. In comparison, cows in their first lactation showed 0.54 mg/dL more MUN concentrations compared to second- and third-lactation cows [28]. This could be due to cows in their first lactation still growing, with lean tissue growth, and having higher efficiency for protein utilization, and thus higher milk urea can be observed [44]. However, in disagreement with our study, other research has found an increase in MUN concentrations with increasing lactation number [39,45] or no differences [46]. The conflicting findings on MUN concentrations and lactation numbers can be due to different nutritional management systems. Since cows in their first lactation are still growing and adapting to lactation, they require more protein [44]; thus, feeding the same diet to higher-lactation-number cows can influence their MUN concentrations [46].

### 4.2. Body Condition, Lying Time, and Lameness

Body condition monitored over time suggests whether a cow has been in a positive or negative energy balance for an extended period [47]. Breed type and parity may also impact body condition, as they directly influence the efficiency and physiological demands of dairy cows. There is a lack of research on the effect of breed on BCS. However, in comparison to our study, others have found that Holsteins have a lower BCS than Jerseys [48] and that Holsteins and crossbreeds have similar BCS throughout lactation [31]. Additionally, the current research found that BCS decreases with increasing lactation numbers. In comparison, other research has found that cows in their fourth lactation had a delayed recovery of body condition until 90 d in lactation, which could be due to negative energy balance during this period [35]. Research has also found that multiparous cows lose more body condition than primiparous cows up to 100 DIM [49]. This could be due to later-lactation cows being older and primiparous cows having intense energy utilization for development during lactation. For example, Holsteins in early lactation (0 to 100 DIM) had a BCS of 3, increasing BCS during mid-lactation (100 to 200 DIM), and nearly a steady state BCS of 3.5 during late lactation [49]. However, if the cow is unable to restore and maintain a higher body condition throughout lactation, the cow may be in a negative energy balance during most of their lactation.

Lameness in dairy cattle is attributed to many factors. This study focused on the effects of breed and lactation number on lameness scores and how lameness may impact lying time, milk production, and reproductive performance. In our study, we found Jerseys had the lowest lameness scores compared to Holsteins, which agrees with another study that found lameness occurred less frequently in Jerseys compared to Holsteins [50]. However, further research is needed when comparing lameness scores across crossbreeds. In our study, we found that as lactation number increases, so does lameness scores, which agrees with other studies [31,37,38]. The findings that lameness increases with lactation number could be due to observations that as parity increases, so do the chances of developing claw disorders [38].

In our study, for cows that were able to conceive, milk yield had a positive correlation with lameness, suggesting that as milk yield increases, so do lameness scores. This could be because Holsteins and crossbreeds had higher lameness scores along with higher milk production. Thus, the higher-producing breeds were more likely to be lame. In contrast to our findings, other researchers have found lameness to negatively impact milk yield [7,51]. Within the cows that conceived, there was a negative correlation between lameness and MUN concentrations. This suggests that as lameness scores increase, MUN concentrations will decrease, or vice versa. We suggest that this is because MUN concentrations and milk yield had a negative correlation, and thus if milk yield increases (along with lameness), MUN concentration will decrease (along with lameness increasing). Another reason for a negative correlation could be due to lame cows lying down longer and eating less, and thus a decrease in protein intake can cause a decrease in MUN concentrations. Therefore, further evaluation of lying time will contribute to our understanding of the relationship between lameness and MUN concentrations.

In contrast to other studies, we found that lameness did not affect lying time. Other studies found that with increasing lameness scores, there was an increase in lying time [4–6]. In comparison to our findings that Holsteins have an average daily lying time of $8.5 \pm 2.5$ h/d, other studies have found the average lying time of Holsteins to be between $10.6 \pm 2.3$–$11.6 \pm 2.07$ h/d [13,15,52]. The differences in lying time for Holsteins could be based on individual cow factors that have been previously discussed (milk yield, BCS, lameness, TMR quality, lactation number) or due to the management decisions of the farm, including housing and stall design. Holsteins are the largest breed and the highest milk-producing breed; thus, Holsteins are more likely to have a greater DMI than smaller breeds that produce less volume, such as Jerseys, to maintain milk production. Therefore, Holsteins could be seen to take precedence for feeding over lying time. Due to a low BCS observed for Holsteins, we suggest that this could cause Holsteins to be in negative energy balance; thus, they would consume more feed to increase the energy supply for their high milk yield. Additionally, more research is needed to compare the lying times of Jerseys and crossbreeds.

In our study, the findings that lactation number did not affect lying time agrees with other studies [52,53]. However, differences in lying time between lactation numbers has been found in other studies. For example, a study found that multiparous cows had longer lying time during early lactation compared to multiparous cows [13], which could be due to multiparous cows being at higher risk of lameness. An increase in lying was seen in cows with higher parity in other studies [14,27]. However, in our study, we found cows in their fifth lactation to have a longer lying time compared to cows in other lactation numbers. We suggest this could be due to the higher lameness prevalence in higher lactation numbers that we saw in our study.

### 4.3. Reproduction

For all cattle, Jerseys had a mean $P_4$ concentration of $9.5 \pm 19.46$ ng/mL and Holsteins had a mean $P_4$ concentration of $6.1 \pm 7.723$ ng/mL from 60 to 130 DIM. There is a lack of research regarding $P_4$ concentrations within breeds from 60 to 100 DIM. Based on the results from our study, it appears that the difference in $P_4$ concentrations among breeds could be due to Jerseys producing less milk, suggesting a lower DMI, and allowing an increase in peripheral concentrations of $P_4$ [54]. However, more research needs to be conducted on $P_4$ concentrations between breeds throughout lactation. Maraes et al. (2016) found Jerseys at $34 \pm 3$ DIM had a $P_4$ concentration of $3.36 \pm 0.94$ ng/mL [55]. The differences in these findings could be because our $P_4$ concentration was averaged throughout the study (average 100 DIM), whereas Moraes et al. (2016) took samples before the end of the voluntary waiting period. Thus, a higher $P_4$ concentration can be seen in later DIM. Brown et al. (2012) found that Holsteins made up $42.2 \pm 5.4\%$ of cows with $P_4$ concentrations increasing >1 ng/mL in the first 30 DIM compared with $43.9 \pm 6.6\%$ of Holstein–Jersey crossbreeds, $61.7 \pm 5.4\%$ of Jersey–Holstein crossbreeds, and $67.4 \pm 7.2\%$ of Jerseys [56].

These findings suggest that $P_4$ concentrations can continue to increase throughout lactation, supporting our results of higher $P_4$ concentrations in a longer DIM.

In our study, cows in their second lactation had a $P_4$ concentration of $11.772 \pm 20.3589$ ng/mL, which was greater than other lactation numbers. However, another study found that cows in their second lactation had a maximum $P_4$ concentration of $2.5 \pm 2.2$ ng/mL by d 72 through d 87 [57]. The differences in $P_4$ concentrations could be because cows in their second lactation in our study had a lower $P_4$ concentration coming into 60 DIM and needing more prostaglandin treatments to stimulate hormonal activity, thus increasing $P_4$. In comparison to our findings, Walters et al. (2002) found that serum $P_4$ declined by d 87 postpartum in third-lactation cows [57]. However, primiparous cows had a 5.35% higher $P_4$ concentration than multiparous cows from day 35 to insemination [54]. Therefore, cows in early lactation with high $P_4$ concentrations could be expected, as a higher milk yield in multiparous cows would be associated with greater dry matter intake and consequently accelerated metabolic clearance of $P_4$, decreasing its peripheral concentrations [54].

Lameness did not affect the CCI in our study. However, other studies have found lameness to increase the calving-to-conception interval by 13 to 14 days [58], increase the time-to-conception interval by 66 days [59], and lengthen the calving interval [10]. Our differences could be due to lower lameness scores in the conceived cows within our study, whereas other studies included cows with higher levels of lameness.

In cows that conceived during this trial, the correlation between $P_4$ concentrations and CCI tended to be positive. This suggests that as $P_4$ concentrations increased, so did the CCI and vice versa. Researchers suggest that this correlation could be due to cows not getting inseminated at the time of estrus, and thus cows would return to the luteal phase of their estrous cycle. During the luteal phase, as the CL increases in size, there is a rise in secretion of $P_4$ [60]. However, further research is needed to compare the relationship between $P_4$ concentrations and the CCI.

The relationship between $P_4$ concentrations and THI had a significant negative correlation, suggesting that as the THI increases, $P_4$ concentrations decrease, which could be due to heat stress impacting hormonal regulation [61], including cyclicity and estrus expression. Researchers have found a decrease in ovarian cyclicity (identified through low $P_4$ concentrations) with an increase in THI [62]. The relationship between the CCI and THI tended to have a negative correlation, suggesting that as the THI increased, there was a tendency that the CCI decreased. However, our findings disagree with other research. High THI was associated with decreased cyclicity and estrous expression [3,25], thus increasing the probability of estrus detection failures [63]. A high THI value, indicative of heat stress, can have negative impacts on conception rates [36,64]. Therefore, it is more likely to see an increase in the CCI and other reproductive measures including $P_4$ concentrations due to the negative effects of high THI.

## 5. Conclusions

This study evaluated different cow parameters and their effect on milk yield, MUN, $P_4$ concentrations, CCI, lying time, and lameness. Differences were found between breeds and lactation numbers in relation to lameness, milk yield, MUN concentrations, BCS, lying time, and reproductive measures. Jerseys produced the least milk, had the highest BCS, MUN, $P_4$ concentration, and lying time, and the lowest lameness scores compared to Holsteins and crossbreeds. Cows in their first and fifth lactations produced the least milk, whereas those in lactation numbers one and two had the highest MUN concentrations. As lactation numbers increased, so did lameness scores, but with a decrease in BCS. Further research is needed to fully understand the effects of cow parameters (breed and lactation number) on the behavior and the calving-to-conception interval. It is suggested that cow-based factors including breed type and lactation number play a bigger role in different productivity aspects. Management decisions also play a big role in maintaining certain breed types, lactation numbers, and productivity of cows. The management of lameness

in dairy cattle in Texas will benefit by focusing on the effects of breeds, lactation numbers, and associated management decisions to reduce lying time and improve reproduction and milk production.

**Author Contributions:** Conceptualization, L.A.M., E.C.W., B.W.J., J.A.S. and C.L.R.; methodology, L.A.M.; formal analysis, E.C.W.; investigation, L.A.M.; resources, B.W.J. and L.A.M.; data curation, L.A.M.; writing—original draft preparation, L.A.M.; writing—review and editing, E.C.W., J.A.S., B.W.J., C.L.R. and K.B.W.; visualization, L.A.M., B.W.J. and E.C.W.; supervision, E.C.W. and project administration, L.A.M.; funding acquisition, B.W.J. All authors have read and agreed to the published version of the manuscript.

**Funding:** Research was funded by the National Institute of Food and Agriculture (NIFA), grant 2021-70001-34521.

**Institutional Review Board Statement:** The animal study protocol was approved by the Institutional Animal Care and Use Committee (IACUC) of Tarleton State University (IACUC approval number: 05-022-2023).

**Informed Consent Statement:** Not applicable.

**Data Availability Statement:** Dataset available on request from the authors.

**Acknowledgments:** The authors thank the farm owner and staff who allowed us to conduct research on their farm. The authors also thank the graduate students who assisted with data collection. The first author expresses gratitude and appreciation towards the faculty of Tarleton State University for their support.

**Conflicts of Interest:** The authors declare no conflicts of interest.

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
