# Peer review of "The Effects of Breed, Lactation Number, and Lameness on the Behavior, Production, and Reproduction of Lactating Dairy Cows in Central Texas"

_ruminants, doi:10.3390/ruminants4030023_

Round 1

Reviewer 1 Report

Comments and Suggestions for Authors

Dear authors,

Congratulations with your very thorough work. The M&M section as well as the results and discussion sections are very detailed. and the easiest read. I suggest you work a bit on the conclusion, so that it is clearer to the reader what the results can be used for. 

I have some other minor suggestions.

In general: To help the reader, make sure that every time you use an abbreviation the word/term is written in full length.

In your introduction, I miss a description of the usefulness of MUN and progesterone concentration - if you include this in this section, it is easier to understand later in the text, why these measurements are included in the study.

Line 61: "multiplied by... "something is missing here.

Line 102: What kind of footbath?

Line 120: What defines "sick"? Is that everything that isn't lameness or mastitis?

Author Response

Dear sir / madam,

Thank you for the constructive criticism. We have attended to all the reviewer comments, and I attach a detailed description of all the comments and our responses. (Please see attached the rebuttal and corrected manuscript in which the edits are highlighting).

Reviewer 2 Report

Comments and Suggestions for Authors

Dear authors, the work seems to be written in an interesting way, although it is not groundbreaking.

First of all, the research was carried out on animals, and blood was also collected from animals, yet the consent of the ethics committee was not mentioned.

In addition, the text requires some changes:

1. line 25 - no explanation of the abbreviation DIM - and it will appear here for the first time

2. line 61 - "multiplied by 100

3. line 83 - methodology is unclear. Since the work assumes a comparison between breeds, too few animals were selected, which the authors later try to hide by providing average values ​​for all animals in the results

4. line 98 - remove the dot between "and measured 2m..."

5. line 104 - add a colon at the given times: it should be 09:00 AM to 05:00 PM

6. line 137 - expand the description of the lameness rating scale

7. line 157 - unfinished sentence: "(...) was used for the analysis of categorical data namely BCS and ???."

8. line 162 - incorrect presentation of results in tables. There are no differences between breeds in individual lactations, leading to incorrect conclusions. All values ​​were averaged

9. line 185 - justify the subtitle

10. line 200 - what are the parameters of the type: "493.427 (14, N=6457) = 493.427". Similar phrases appear in every paragraph of the results. They are not included in the tables - they are incomprehensible to the reader

11. line 205 - add a space before the bracket, it should be: "179.806 (6"

12. line 208 - add a space before the bracket, it should be: "1847.6 (18")

13. line 208 - in fact, only at lactation 4 an increase in lameness was observed - sentence to be reworded

12. line 223 - the results of the mentioned correlation between lameness and milk yield are not provided in the suggested Table 3

13 line 443 - should be: "big role"

14. The conclusion should be changed. The obtained research results cannot be cited again. It is intended to be a clear and transparent conclusion summarizing all the research and at the same time giving importance to the research conducted. Moreover, as mentioned earlier, the methodology adopted by the authors and the way of presenting the results in tables do not justify the conclusions drawn regarding interracial differences.

15. the list of literature should standardize whether "doi" or "Doi"

Author Response

(The authors gave the same response as above.)

Reviewer 3 Report

Comments and Suggestions for Authors

This study evaluated the impacts of breed, lactation number (parity) and lameness on multiple production parameters in dairy cows in Texas. The paper adds to the overall knowledge base regarding cow characteristics which might influence production parameters for dairy cattle, especially within the U.S.A. The study looked at many items and utilized a commercial dairy facility for data, making the data very applicable to producers and not just in a controlled research setting. I commend the authors for this approach and for being able to manage the intricacies of conducting research at a non-research farm given the multitude of hurdles which can be present doing so. While this study has some good application, there are a few questions I have with it.

General Questions / Comments:

- Be sure to define abbreviations the first time they are used within the actual paper text, not just relying on utilization from the abstract.

- Do you believe that you have enough data variation for the lactation number analysis, especially 1st lactation cows, to be able to tease out those affects fully as described in the paper?

- Were there any significant interactions present? There is no mention of these results, which could impact the data presented.

- How many inseminations, per cow, on the group which "did not concieve" for this study and did you follow up with them to identify the total number of inseminations to produce a pregnancy?

- How were cows classified as "no significant structure" or having a "CL less than 15 mm" handled and data reported?

- You measured THI, but I don't recall seeing any results relating to it; was there an impact of it on any parameters measured?

- Many correlations discussed in the results and discussion are not presented in the current tables. These data would probably be helpful to present in a table or two within the manuscript since they are fairly paramount to the discussion and impacts of the study.

- For the tables, the superscripts are confusing, as I am curious if you are comparing the data from Breed against the data from Lactation number to obtain some of the differences. Also, the tables need to be able to be "stand-alone", i.e. terms defined with footnotes where necessary and fully self-explanatory.

Specific Comments

- Line 52: close parenthesis after "16 - 18"

- Line 60: change "at" to "as"

- Line 61: "multiplied by" is missing a number after it, likely 100

- Line 79 & 80: make the effects singular instead of plural

- Line 104 & 105: time should be "0900 to 1700 h" and ad libitum should be italicized

- Line 114: change "Rahway" to "Rathway"

- Line 157: "BCS and" is missing a word or words

- In Table 1: Lactation Number 4 & 5 the superscripts for MUN & Milk yield should be double-checked

- Line 189: double check the statement about milk yield differences between breeds

- In Table 2, 4, & 5: for lactation number 1 is this data from 1 cow? If so, how much validity is there in using this data?

- Lines 198 - 209: Based upon the data provided in Table 3, this section makes no sense as the numbers presented do not correspond with anything in the table

- In Table 2, 4, & 5: the "correlation & p-value" at the bottom of the table is confusing as to what exactly you are trying to present. Per my comment in the general comments section, I would suggest adding a new table with the correlation values of interest for clarity

- Line 389: is VWF defined previously?

Comments on the Quality of English Language

Only a few minor grammatical issues

Author Response

(The authors gave the same response as above.)

Round 2

Reviewer 3 Report

Comments and Suggestions for Authors

I appreciate the authors revisions and comments regarding the 1st review, though a few items seem to still be unaddressed.

- Line 110: 9 am & 5 pm should be 0900 h & 1700 h

- LInes 146 - 151: add discussion regarding insemination rate or number of insemination before excluding cattle from the data and considering them to "not conceive"

- In results: Were interactions present? If not, this should be addressed as it was mentioned in your statistical analysis of Materials & Methods

- In results: I am skeptical of describing differences between Lactation 1 and any other lactation due to the lack of cows present, even if using repeated measures as you are limited in the animal variability when only 1 first lactation cow conceives. If your data is based on repeated measures, why is table 2, 4, & 5 have SEM values for lactation 1 or 0?

- Line 205 - 216: The data in these 2 paragraphs is incomprehensible to the reader even given that these are for categorical variables.

Comments on the Quality of English Language

No issues

Author Response

Thank you for these comments and corrections - we have addressed the comments as follows:

Comment 1: Line 110: 9 am & 5 pm should be 0900 h & 1700 h

Response 1: Completed and corrected as suggested.

Comment 2: LInes 146 - 151: add discussion regarding insemination rate or number of insemination before excluding cattle from the data and considering them to "not conceive"

Response 2: Lines 115- 121 includes veterinarian procedures for determining conception and in the cows. Lines 127 – 128 in “data collection” states there were no exclusions based on number of services or health events.

Comment 3: In results: Were interactions present? If not, this should be addressed as it was mentioned in your statistical analysis of Materials & Methods

Response 3: Added that no interactions were present in the results (line 250-251).

Comment 4: In results: I am skeptical of describing differences between Lactation 1 and any other lactation due to the lack of cows present, even if using repeated measures as you are limited in the animal variability when only 1 first lactation cow conceives. If your data is based on repeated measures, why is table 2, 4, & 5 have SEM values for lactation 1 or 0?

Response 4: Lactation #1 is now excluded from tables 2, 4, and 5 since the data is repeated measures from one cow.  

Comment 5: - Line 205 - 216: The data in these 2 paragraphs is incomprehensible to the reader even given that these are for categorical variables.

Response 5: Lines 205 – 216 were edited completely to reflect the correct effects with necessary p-values. The direction of the effects was also explained.

Round 3

Reviewer 3 Report

Comments and Suggestions for Authors

I thank the authors for their revisions and manuscript changes.

There are a few minor issues during this latest review

-Line 206: Starting the sentence with BCS is not ideal since it is an abbreviation and I believe that there may have been some wording lost during revision.

- Line 208 & 209: I am unsure what is meant by "1155.056 (42, N=6457) = 1155.056"?

- Line 215: Starting the sentence with "(p<0.001)" is not ideal and I believe some extra wording was removed during revision.

Comments on the Quality of English Language

No major issues found

Author Response

Dear editor,

Thank you for these important editorial corrections. These errors were due to the previous revision. 

Comment 1: -Line 206: Starting the sentence with BCS is not ideal since it is an abbreviation and I believe that there may have been some wording lost during revision.

Response: This has been corrected by giving to correct description, e.g. 'Body condition score (BCS)'.

Comment 2: - Line 208 & 209: I am unsure what is meant by "1155.056 (42, N=6457) = 1155.056"?

Response: The comment is correct, namely that these statistics are not necessary, so it was deleted.

Comment 3: - Line 215: Starting the sentence with "(p<0.001)" is not ideal and I believe some extra wording was removed during revision.

Response: This was deleted as this incorrect typing was a remnant of the previous editing that slipped through.

Please note that I have edited Line 15, to read better. Please check if this is acceptable.

Please see the corrections in the edited version of the manuscript.

Sincerely,

Edward Webb
